



# Macapá, a Brazilian equatorial magnetometer station: installation, data availability and methods for temperature correction.

Cristiano Mendel Martins[1], Katia Jasbinschek Pinheiro[3], Achim Ohlert[2], Juergen Matzka[2], Marcos Vinicius da Silva[2,3], and Reynerth Pereira da Costa[3]

[1]Geoscience Institute, Universidade Federal do Pará - UFPA, Pará, Brazil.
[2]GFZ German Research Centre for Geosciences, Potsdam, Germany.
[3]Observatório Nacional- MCTI, Geophysics Department, Rio de Janeiro, Brazil.

**Correspondence:** Marcos da Silva (mvsilva@gfz-potsdam.de)

**Abstract.** In the last 60 years, the largest displacement of the magnetic equator (by about 1100 km northwards) occurred in the Brazilian longitudinal sector. The magnetic equator passed by Tatuoca magnetic observatory (TTB) in northern Brazil in 2012 and continues to move northward. Due to the horizontal geomagnetic field geometry at the magnetic equator, enhanced electric currents in the ionosphere are produced - the so called equatorial electrojet (EEJ). The magnetic effect of the EEJ is observed

in the range of $\pm$ 3 degrees from the magnetic equator, where magnetic observatories record an amplified daily variation of the H component. In order to track the spatial and temporal variation of this phenomena, a new magnetometer station was installed in Macapá (MAA), which is about 350 km northwest from TTB. In this paper, we present the setup and data analysis of MAA station from 11/2019 until 09/2021. Because of its special configuration, we develop a method for temperature correction of the vector magnetometer data.

## 1   Introduction

With globally distributed geomagnetic observations it is possible to investigate the different geomagnetic field sources in the core (largest part of the measured field), crust, ionosphere, magnetosphere and induced fields (Hulot et al., 2010). Temporal variations of the Earth's magnetic field at ground are monitored by geomagnetic observatories (Matzka et al., 2010) and magnetometer stations (Chulliat et al., 2017). Dedicated geomagnetic satellite missions provide observations from low-Earth-orbit.

Magnetic observatories produce high-quality and continuous data over long periods (Matzka et al., 2010). Precise and frequent absolute measurements of declination and inclination by trained staff are required to calibrate geomagnetic observatory data. Observatories belonging to INTERMAGNET (International Real-time Magnetic Observatory Network) follow quality standards for measuring and transmitting real-time data (St-Louis et al., 2020). They are an important data source for studies of the internal geomagnetic field and its secular variation as well as studies of space climate, e.g. with the help of geomagnetic

indices. However, there is an uneven spatial distribution of magnetic observatories around the globe, which is worse in the Southern Hemisphere and oceans. There are many reasons for such uneven distribution, such as infrastructure requirements, data transmission problems (especially in remote areas), need of trained staff and lack of investments that are fundamental to maintain or construct new observatories. On the other hand, magnetometer stations are not so demanding in terms of infras-





tructure and staff available. The three components of the magnetic field are measured continuously in a magnetometer station,
such as it is done in the observatories, however, absolute measurements are not periodically acquired. Typically, magnetometer
stations are intended to monitor the external and induced geomagnetic fields. They can also be used as a first step to investigate
the suitability of a new location before constructing a geomagnetic observatory.

In Brazil, there are two INTERMAGNET observatories: one in Vassouras (VSS - Rio de Janeiro, RJ) and another in Tatuoca
(TTB - island in Belém, PA), which measure continuously the magnetic field since 1915 and 1957, respectively. There are two
important magnetic phenomena in Brazil: the South Atlantic Anomaly (SAA) where the magnetic field intensity is the smallest
in the globe and the magnetic equator, where the magnetic inclination is zero.

In the last 60 years, the displacement of the magnetic equator in Brazil was the largest in the entire globe: 1100 km north-
wards, as predicted by the IGRF-13 (International Geomagnetic Reference Field) model (Alken et al., 2021). In figure 1 the
magnetic equator is shown for different epochs in the last 60 years and the forecast for 2024, where the secular variation of
the inclination predicts the largest changes over Brazil (Alken et al., 2021). In this zero-inclination region, the magnetic field
is mostly horizontal and, as a consequence, strong ionospheric electric currents at about 105 km altitude are produced on the
day-side. These currents extend to about $\pm3^0$ from the magnetic equator and are known as the Equatorial Electrojet (EEJ),
as reviewed in Yamazaki and Maute (2017). The result is an intensification of the H-component daily variation (Soares et al.,
2020), reaching up to about a hundred nanoteslas (see example in Fig. 2). In 2012 the magnetic equator passed over TTB
and its amplified diurnal variation was analysed by Soares et al. (2020). When compared to fields generated by solar quiet
currents (Sq) from low and medium latitudes, the magnetic field in TTB shows a special variation, influenced by the seasons,
atmospheric oscillations and lunar tides.

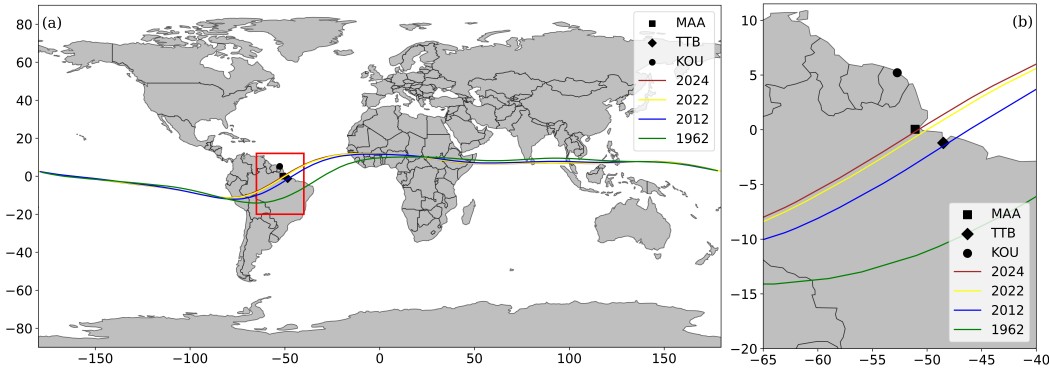

**Figure 1.** Magnetic equator calculated by the IGRF-13 (a) for the 1st July of the following years: 1962 (green line), 2012 (blue line), 2022 (yellow line) and 2024 (red line). The zero inclination lines are calculated considering 105 km altitude, region where the equatorial electrojet is produced. In (b) is shown a zoom of South America, with the locations of Tatuoca (diamond), Kourou (circle) and Macapá (square).

The magnetic equator continues to move northwards and it is predicted by the IGRF-13 model (Alken et al., 2021) to be
located at Macapá station at around July 2024 at Macapá station altitude (see Fig. 1). Therefore, a new magnetometer station
was installed in Macapá in order to track the effects of the Equatorial Electrojet. This work is a result of a cooperation between





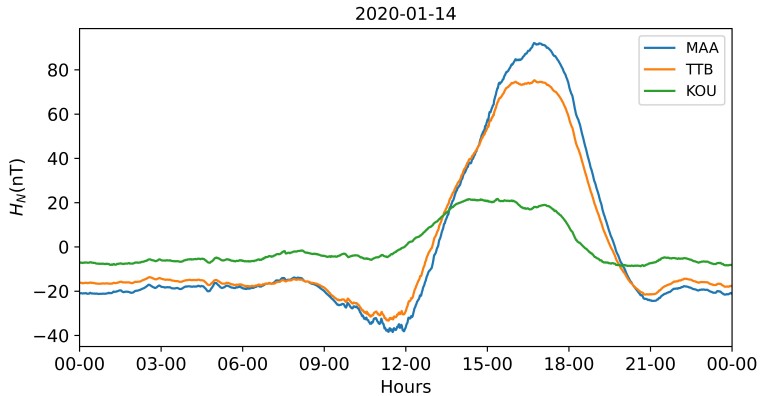

**Figure 2.** Comparison between the north component daily variation of Tatuoca (TTB, 1.20°S, 48.51°W), Kourou (KOU, 5.21°N, 52.73°W) Magnetic Observatories and north-component in the sensor coordinate system ($H_N$) of Macapá station on the 14th January, 2020. Local noon at TTB is 15:00 UTC.

Observatório Nacional (Rio de Janeiro, Brazil), Universidade Federal do Pará (UFPA, Brazil) and GFZ Potsdam (German Center for Research in Geosciences, Germany).

In order to choose the best location for Macapá station, two sites were tested in May 2019: at Chaves (0.180°N, 49.954°W and 2m altitude) and at IEPA (Institute for Scientific and Technological Research of the State of Amapá, 0.038°N, 51.09°W, 34 m altitude). The total magnetic field was measured at both locations as well as 50 meters to the North, South, East and West of it. Since just a weak magnetic gradient of around 1 nT was found, the total field was measured for three consecutive days in both locations in order to check the data quality and possible noise from environmental interference. Both locations showed good data quality and good agreement with simultaneous TTB data. The access to Chaves is more difficult due to a 8-hour trip by boat, while IEPA is more accessible. In addition, since IEPA is a public institution, local staff and infrastructure are available, which are both fundamental for the success of magnetometer stations in remote areas.

## 2   Macapá Station Setup

In November 2019 the magnetometer station of Macapá (MAA) was installed in the IEPA campus (Fig. 3). There is no unique technique to install a magnetometer station, since each location has its own requirements depending on the local conditions like whether, infrastructure and staff availability. Two instruments measuring the magnetic field were installed in Macapá. Two instruments measuring the magnetic field were installed in Macapá. A triaxial fluxgate magnetometer (GEOMAG-02 by Research Centre GEOMAGNET) measures in the sensor coordinate orthogonal system the horizontal north ($H_N$), horizontal east ($H_E$) and the vertical ($Z$) components (Table 1). On the installation of the GEOMAG sensor set, we align it in a way that $H_E$ is zero. The sampling rate is 1 Hz and the time stamping is controlled by GPS and Network Time Protocol (NTP). The





second instrument is a scalar overhauser magnetometer (GSM-90 overhauser by GemSYS), which measures the scalar field

strength F. The sampling rate is 1 sample per 5 seconds and for the time stamp is used NTP.

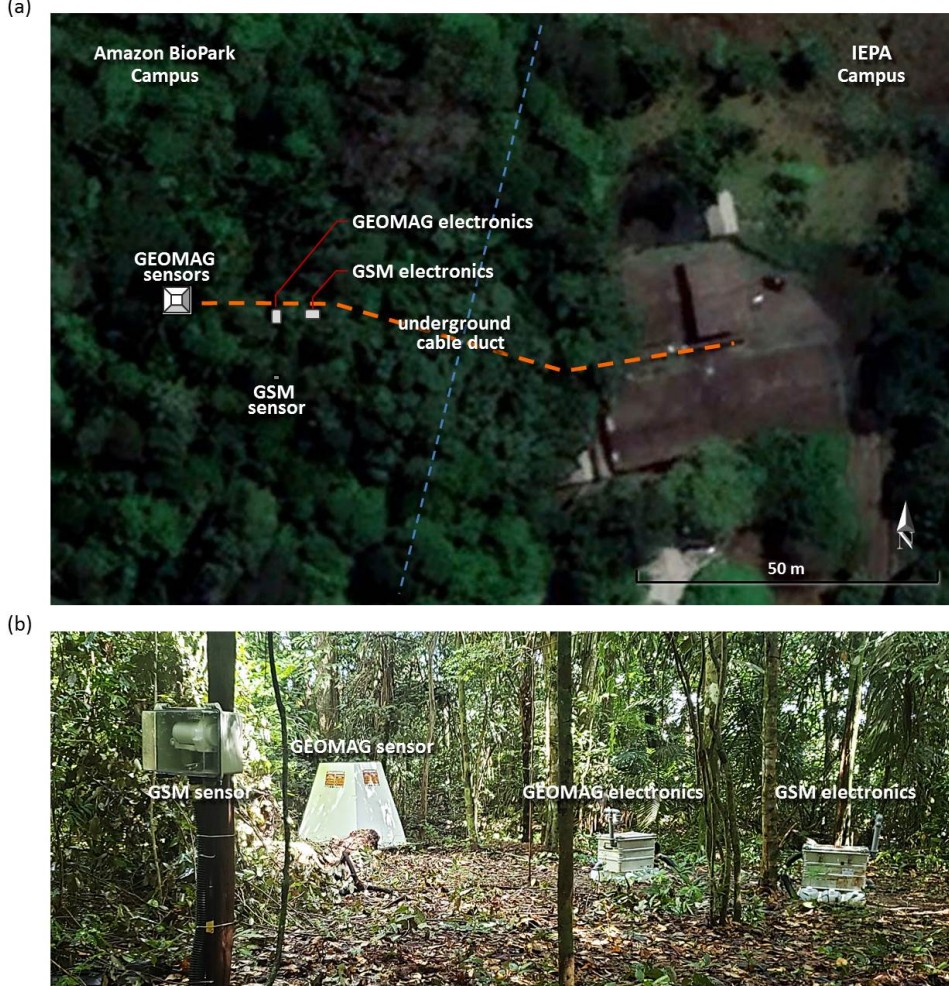

**Figure 3.** (a) Location of Macapá station (IEPA campus) showing the locations of the sensors and electronics (Google Earth image 2021 © Maxar Technologies) and (b) fibreglass pyramid and the boxes with GEOMAG and GSM electronics.

    The GEOMAG sensor is installed inside a fibreglass shelter for protection against rain and wind (Fig. 3). This shelter has thermal insulation and is pyramid-shaped (with a flat top, see Fig. 4), as it was done for Tristan da Cunha observatory Matzka et al. (2011). This design is similar but lighter than the pyramid described in Matzka et al. (2011). The GEOMAG electronics is in a box about 5 meters from the pyramid (Fig. 4). The pyramid is located in the forest, around 70 m from the office building.

Local staff assured that in this region there is no flood in the rainy season and no fires during the dry season. To the east of the pyramid, there are the buildings of the IEPA and to the west, the "BioParque Amazonia" that is a newly opened attraction park/zoo. To the south, a larger street can be found at about 300m distance, with a forest in between.





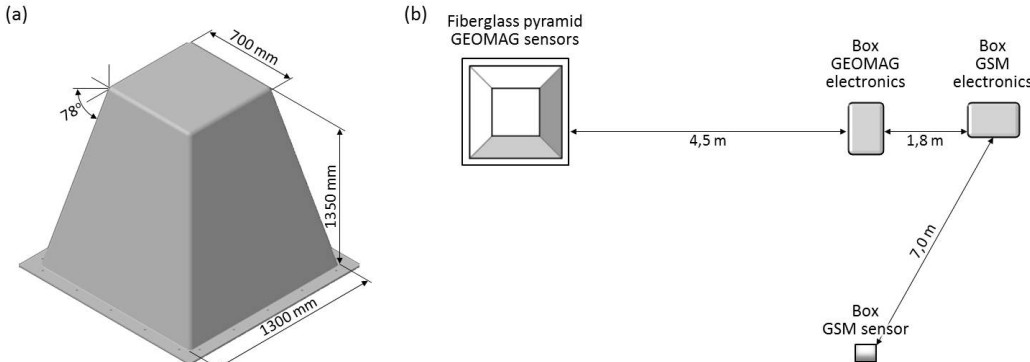

**Figure 4.** Fibreglass pyramid dimensions constructed for Macapá magnetometer station (a) and schematic locations of the fiberglass pyramid with GEOMAG sensor, boxes with GEOMAG and GSM electronics and box with GSM sensor (b).

## 3   Macapá Dataset

Data from Macapá station (MAA) started to be recorded on 14th November 2019 until 25th July 2022 with an availability of
93% for scalar data and 35% for the vector data. The time distribution of MAA dataset is shown in Fig. 5.

Since GEOMAG stopped to transmit data automatically via its serial RS-232 port on the 5th December 2019, the data needed to be transferred by exchanging CF cards once a week. From this time on, the recording by the vector magnetometer had several interruptions due to technical problems such as damage in the GPS. Because of high temperatures in Macapá and insufficient shading of the electronics box, the GEOMAG-02 electronics inside it reached more than 50°C, exceeding its
maximum operating temperature of 40°C recommended by the manufacturer.

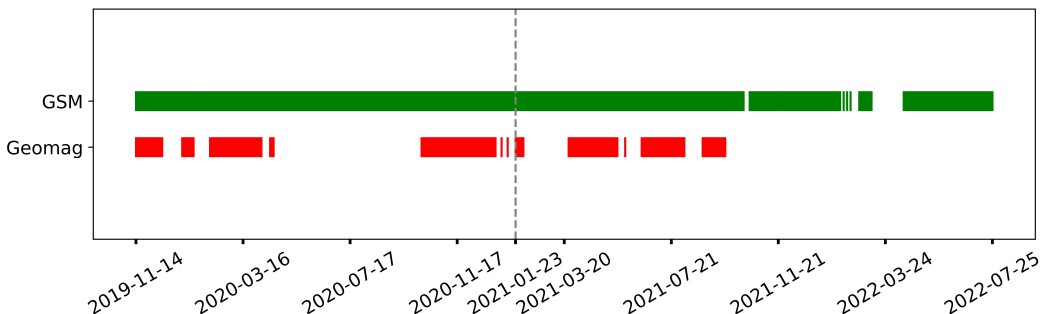

**Figure 5.** Temporal distribution of the scalar (GSM, green) and vector (GEOMAG, red) data available in Macapá magnetometer station. The dashed vertical line indicates the starting time of the synchronization problem.

The GPS of the GEOMAG stopped working on 23rd January 2021. This caused a time shift problem on the vector magnetic data (see example in Fig. 6). Since the GSM scalar data has a correct time stamping, we could apply a correction to synchronize both signals. To do so, we apply this sequence:





[label=.]obtain a first approximation of the delay, by finding for each day the time difference of the maxima in calculated F of the fluxgate and the measured F data of the overhauser systematically shift the time correction for the vector data, starting from the approximate delay found in (i) and calculating the correlation between the shifted calculated F data and the measured F data for each day, selecting the time stamp correction with the highest correlation for the whole period, a least-square fitting by spline to the selected time stamp corrections from (iii) this fitting yields a smooths time error correction for the whole period.

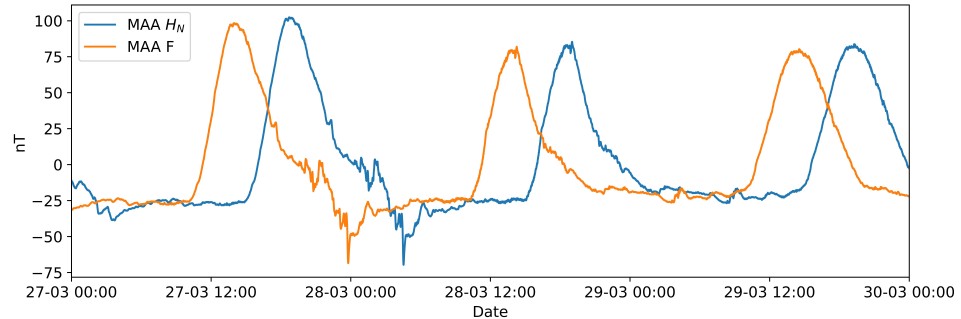

**Figure 6.** Example of a time shift caused by a GPS problem in Macapá station. The total field $F$ was measured by the scalar magnetometer GSM and $H_N$ by the vector magnetometer GEOMAG.

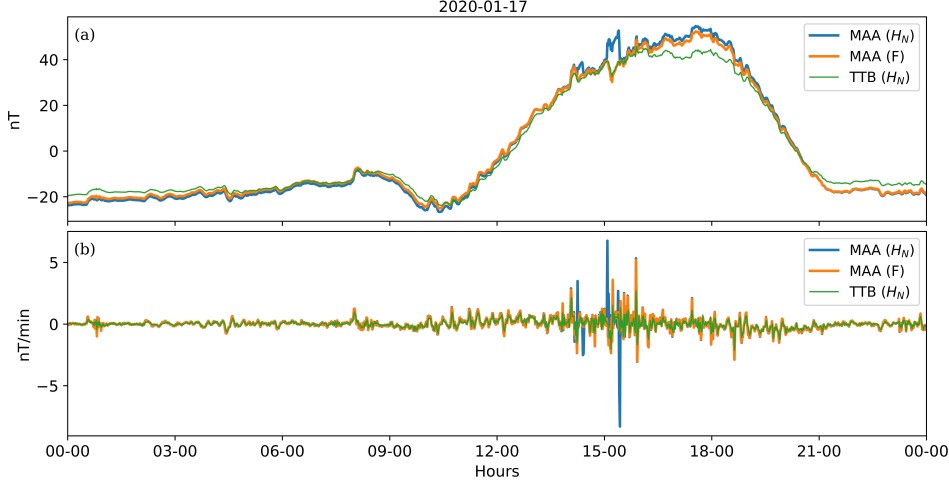

**Figure 7.** Example of artificial disturbance on the 17th January (a) shown in $H_N$ of MAA (blue line), compared to no disturbed signal in $H_N$ of TTB (green line) and total field (F) at MAA (orange line). First time derivative (b) of the $H_N$ of MAA, TTB and F in MAA.





After synchronisation, we removed spikes and artificial disturbances from the data. Spike amplitudes exceeding 1 nT were removed. Spikes were detected by visually checking the first time derivative of each component. Artificial disturbance was visually identified in the geomagnetic data as well as its first time derivative and manually removed (see example in Fig. 7).

After the removal of spikes, MAA data was visually compared to TTB, showing a good agreement of $H_N$ and $H_E$ and a certain degree of anti-correlation in $Z$ (Fig. 8), as is expected from the station location and the geometry of the equatorial

ionospheric current system . Both TTB and MAA are under the EEJ influence (see Fig. 1), presenting a very similar effect of H-component amplification (Fig. 8). Kourou (KOU) is the closest magnetic observatory from TTB but it is still far from the magnetic equator, therefore free from EEJ effect. Similarly as in Morschhauser et al. (2017b) and Soares et al. (2018) it is possible to isolate the localized EEJ effect from large-scale magnetospheric and large-scale solar quiet (Sq) signal by subtracting the data of KOU from the MAA magnetometer station. We calculated the differences between the horizontal

components of MAA and KOU and TTB and KOU (Fig. 9). The differences indicate the intensity of the EEJ signal, which here is around 60 nT (peak to peak). Note that the difference for MAA is slightly greater then for TTB, indicating that the EEJ effect is slightly larger in MAA than in TTB.

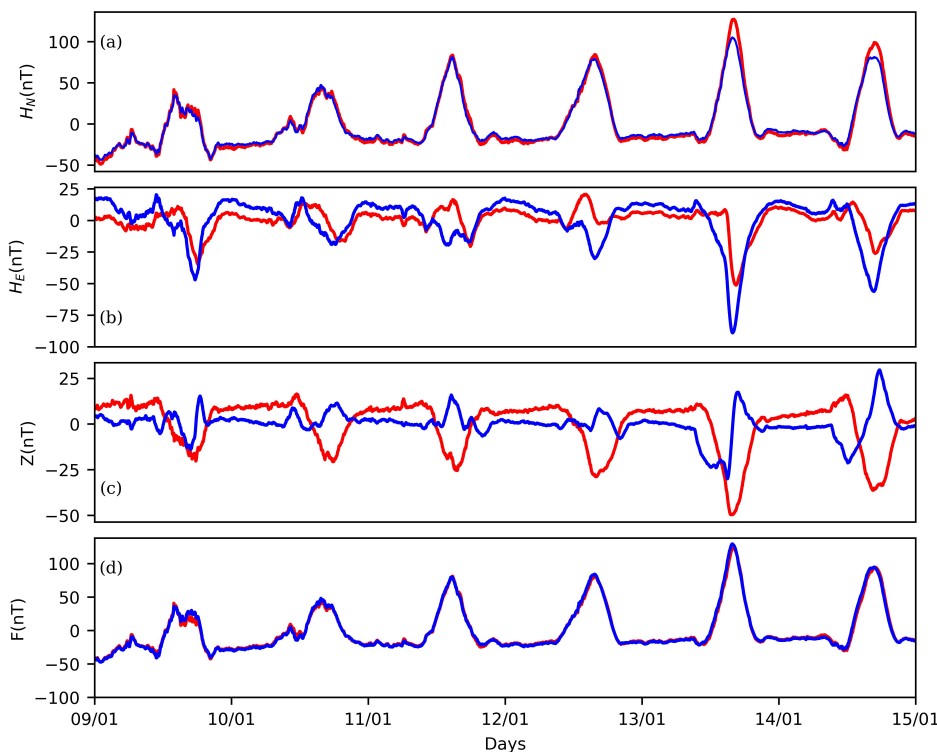

**Figure 8.** Comparison between the the data of TTB (blue) and MAA (red) from the 09th - 14th January, 2020 for the $H_N$ (a), $H_E$ (b), $Z$ (c) and $F$ (d).





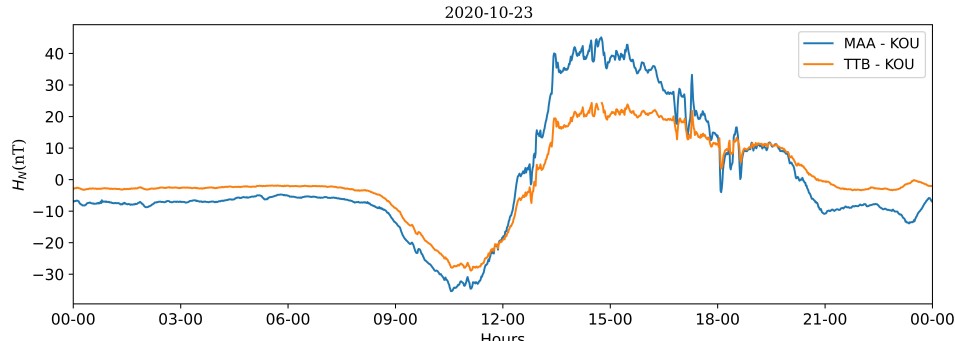

**Figure 9.** Differences betweeen the horizontal components of Macapá (MAA) and Kourou (KOU) (blue line) as well as Tatuoca (TTB) and Kourou (brown line).

## 4 Temperature correction

The total magnetic field is measured directly by the scalar magnetometer (here denoted as $F_s$, measured by GSM-19, Table 1), assumed temperature invariant (Jankowski and Sucksdorff, 1996). It can also be calculated using the three-components recorded by the vector magnetometer (here denoted as $F_v$, measured by the GEOMAG). The temperature is measured in the pyramid (GEOMAG sensor, $T_s$) and in the box (GEOMAG electronics, $T_e$). Hourly mean values $T_a$ of ambient temperature are measured by Instituto Nacional de Meteorologia in less than 2 km distance from the magnetometer station are. All the variable and constant parameters used in this paper are listed in Table 1. The electronic temperature is higher and more variable with time compared to sensor temperature (Fig. 10). They also show a time shift between the temperature peaks, which occur earlier in the electronics temperature. This happens because the electronics produces more heat and it is less insulated from the ambient temperature changes. Figure 10 shows the two temperature signals varying with time.

The calculated total field ($F_v$, Table 1) by using the three vector components is obtained by:

$$F_v(t) = \left[ (H_{N0} + H_N(t))^2 + H_E(t)^2 + (Z_0 + Z(t))^2 \right]^{0.5} \tag{1}$$

where t is time, $H_{N0}$ and $Z_0$ are the sensor offsets (also called baselines) of the north and the vertical sensor (note, the east sensor has no offset). The difference between the calculated ($F_v$) and the observed scalar ($F_s$) total fields is here denoted:

$$\Delta F_0(t) = F_v(t) - F_s(t) \tag{2}$$

and should be zero if both instruments measure correctly. But for Macapá station setup, it varies with a daily periodicity as exemplified in Figure 11a. In order to obtain a possible scaling factor $a_1$ and offset $b_1$ between both signals, we minimize by least squares:

$$min|\Delta F_0(t)| = \left[ \sum_{t=1}^{M} ((a_1 F_v(t) + b_2) - F_s(t))^2 \right]^{0.5} \sim 0 \tag{3}$$





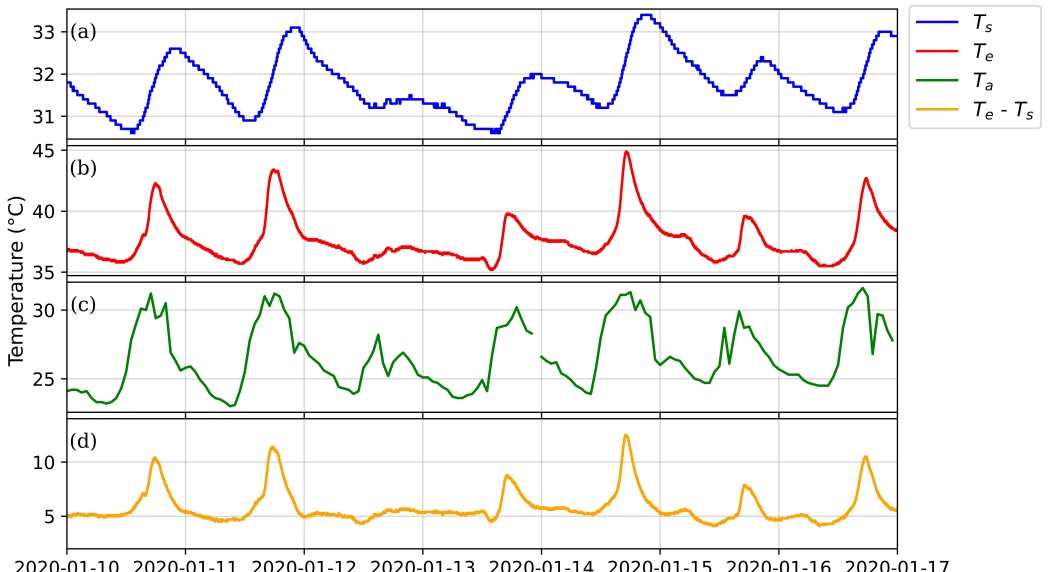

**Figure 10.** Temperature variation of the GEOMAG sensor (a), GEOMAG electronics (b), ambient temperature (c) and the difference between temperature of GEOMAG electronics and sensor (d). The temperature variations range from 10th to 16th January, 2020.

and determine the coefficients by linear regression. We apply this correction to 5-days moving window for all MAA data. We found that $a_1$ values are $0.964 \pm 0.015$ and $b_1$ is negligible ($\sim 10^{-4}$ nT).

$$\Delta F_1(t) = F_{v1}(t) - F_s(t) \tag{4}$$

still contains a periodic signal that is correlated to the sensor temperature ($T_s$), as shown in Figures 11b and 12.

    There is no significant temperature variation between days 12th and 13th January, 2020 (Fig. 10). However, in the same days the $\Delta F_0$ varies with time (Fig. 11a), indicating that there is a diurnal geomagnetic variation besides temperature variation. In order to remove the $T_e$ signal, we minimized $\Delta F_1$ and obtain $a_2$ coefficient by linear regression (Table 1):

$$min|\Delta F_1(t)| = \left[ \sum_{t=1}^{M} \left( (F_{v1}(t) - a_2 T_s(t) + b_2) - F_s(t) \right)^2 \right]^{0.5} \sim 0, \tag{5}$$

where typical values of $a_2$ are $-0.3 \pm 0.15$ nT/°C and $b_2$ is negligible ($\sim 10^{-9}$ nT) to all dataset. We expected that $F_{v2}(t) = F_s(t)$, where $F_{v2}(t) = F_{v1}(t) - a_2 T_s(t)$. However,

$$\Delta F_2(t) = F_{v2}(t) - F_s(t) \tag{6}$$

is still a periodic signal strongly correlated to the difference between the electronic temperature ($T_e$) and sensor temperature ($T_s$), as exemplified in Figure 11. In order to remove the $T_e - T_s$ signal, we minimized $\Delta F_2$ and obtain the $a_3$ coefficient by





linear regression:

$$min|\Delta F_2(t)| = \left[\sum_{t=1}^{M}(F_{v2}(t) - a_3(T_e(t) - T_s(t) + b_3) - F_s(t))^2\right]^{0.5} \sim 0, \tag{7}$$

where $a_3$ typical values are $-0.04 \pm 0.025$ and $b_3$ is also negligible ($\sim 10^{-12}$ nT) for all dataset. Hereupon,

$$\Delta F_3(t) = F_{v3}(t) - F_s(t) \sim 0 \tag{8}$$

where $F_{v3}(t) = F_{v2}(t) - a_3(T_e(t) - T_s(t))$. Finally, the complete expression for the F corrected ($\tilde{F}_v$) is:

$$\tilde{F}_v(t) = a_1 F_v(t) - a_2 T_s(t) - a_3(T_e(t) - T_s(t)) \tag{9}$$

that can be rewritten as:

$$\tilde{F}_v(t) = a_1 F_v(t) - (a_2 - a_3)T_s(t) - a_3 T_e(t), \tag{10}$$

which is the more classical form of describing the temperature dependency of a magnetometer as it assigns a coefficient to the sensor temperature as well as to the electronics temperature. Also INTERMAGNET suggest to monitor both sensor and

electronics temperature and use this correction purposes (St-Louis et al., 2020, p. 10 and 11).

The critical point is that both the sensor and electronics temperature depend on the ambient temperature. Therefore, they are not independent and at the same time they do not have the same shape (Figure 10, A and B). This is because they have different insulation that give a delay in the temperature maximum. After the correction of Ts, we remove part of the dependency on Te, but there is still a signal corresponding to the difference between the temperatures (Figure 10-d) that needs to be removed.

Also note that in the Macapá setup $T_s$ and $T_e$ remain different due to other factors like uneven sun shining on the pyramid and on the box. This occurs because of the different shading by trees (albedo). Another factor, is self-heating of the electronics which is assumed as a constant temperature change. We analyze the correlation between the different temperature signals with $F_s$ to confirm the most appropriate correction sequence, as shown in Figure 12-a. Figure 12-b quantifies how the misfit between $F_s$ and the different $F_v$, decreases with each correction.

## 5   Conclusions

In this paper, we presented the new magnetometer station in Macapá (North Brazil), as a result of the collaboration between the National Observatory (ON - Brazil), the Federal University of Pará (UFPA - Brazil) and the German Research Centre for Geosciences (GFZ - Germany). Macapá station is especially relevant because of the rapid temporal variation of the magnetic equator in the Brazilian longitudinal sector. The magnetic equator passed over Tatuoca observatory (TTB) in 2012 and con-

tinued to move northwards. Today it is located between Macapá station and Tatuoca observatory (Fig. 1). The IGRF model forecasts that the magnetic equator will continue to move northwards and pass by Macapá station in 2024. The presence of the magnetic equator causes another phenomena called equatorial electrojet (EEJ), responsible for an H-component amplification.





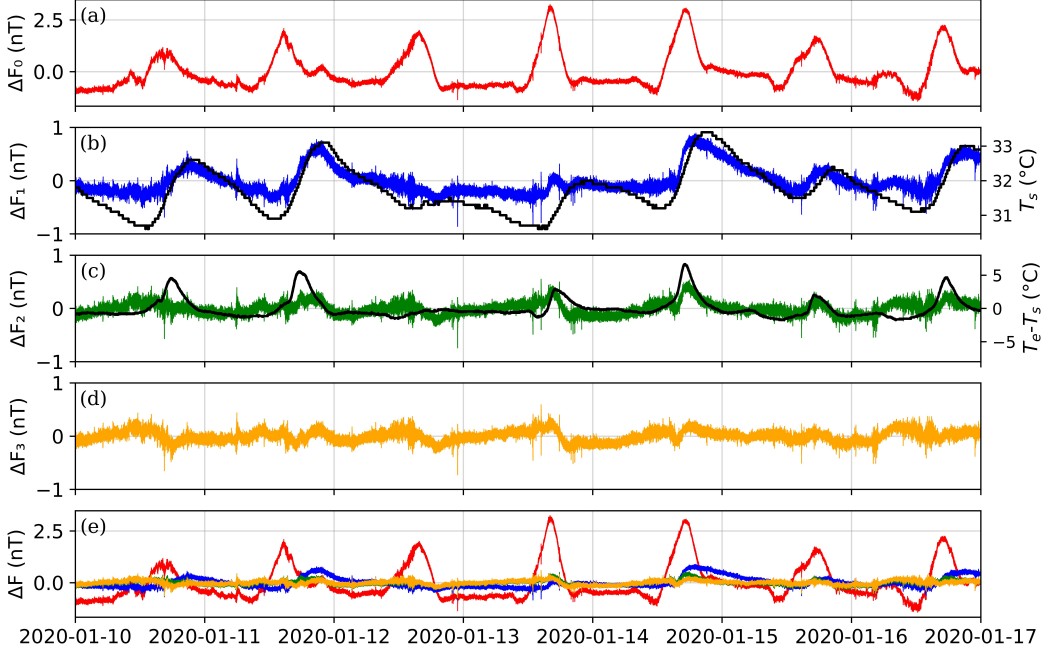

**Figure 11.** Macapá magnetometer station differences ($\Delta F_{v0-3}$) between the calculated total fields ($F_{v0-3}$) and the scalar total field ($F_s$), from $a - d$ respectively. In (b) and (c) the left Y-axis correspond to $\Delta F_v$ and the right scales to the sensor temperature (in b) and to the difference of the electronics and sensor temperatures (in c). All the $\Delta F_v$ are plotted together in (e). The period shown in this example is from 10th to 17th January, 2020.

Since Kourou magnetic observatory is out of the influence of the EEJ, it does not record the H-amplification, as it is observed in Tatuoca magnetic observatory (Fig. 2) and Macapá station.

In Macapá station we measured the total magnetic field (F) with a scalar magnetometer (GSM) and the three-components ($H_N$, $H_E$ and $Z$) with a vector magnetometer (GEOMAG). The sensor and electronics for each magnetometer were installed in different huts (Fig. 3). The GEOMAG sensor was inside a fibreglass pyramid about 5 meters from the electronics box (Fig. 4). We recorded data from Macapá station from 11/2019 until 09/2021 (Fig. 5). There were some problems in the data acquisition, as a GPS failure that caused a time shift between the data of GEOMAG and GSM (Fig. 6). The time shift and other problems

in the data, such as noise and spikes (example in Fig. 7), were corrected. Macapá data was then compared with TTB, which is the closest observatory. The data of TTB and Macapá showed good agreement (Fig. 8), which assure about the quality of the recorded data.

In order to measure the amplitude of the equatorial electrojet (EEJ), we subtract the data from Macapá station and TTB, both under the influence of the EEJ, from Kourou observatory (Fig. 9). The EEJ signal was recorded in Macapá with a similar

amplitude as in TTB.





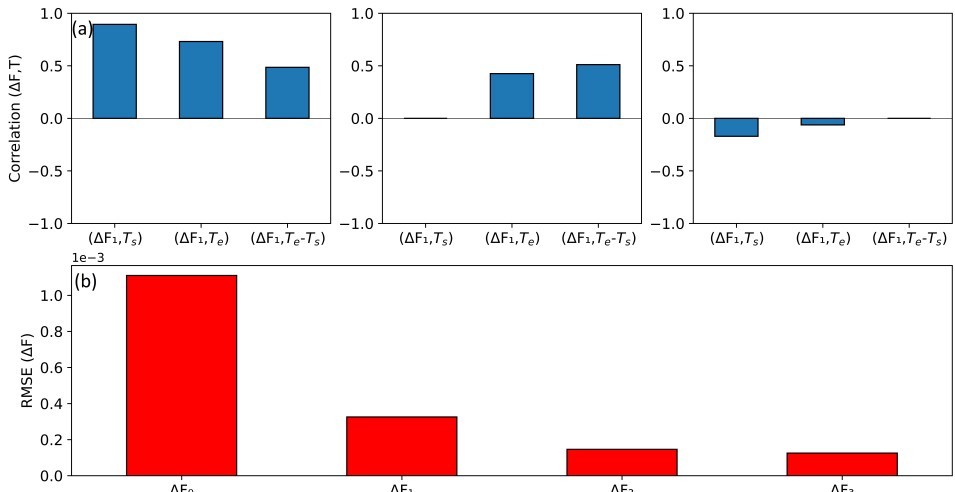

**Figure 12.** Correlation between the temperatures of sensor, electronics and the difference between them and $\Delta F_s$ (blue) and RMSE (Root Mean Square Error) for each $\Delta F$ (red).

**Table 1.** Notation used in this paper for each variable and constant parameters.

| Notation | Description |
|---|---|
| $F_s(t)$ | Observed scalar total field recorded by the GSM. |
| $H_N(t)$ | Horizontal north magnetic field in the sensor coordinate system. |
| $H_E(t)$ | Horizontal east magnetic field in the sensor coordinate system. |
| $Z(t)$ | Vertical magnetic field |
| $T_e(t)$ | Electronic temperature measured inside the box (GEOMAG). |
| $T_s(t)$ | Sensor temperature measured inside the pyramid (GEOMAG). |
| $T_a(t)$ | Ambient temperature of Macapá station. |
| $F_v(t)$ | Calculated total field by using the three vector components of GEOMAG. |
| $F_{v_1}(t)$ | $F_v(t)$ corrected by linear regression using $F_s(t)$. |
| $F_{v_2}(t)$ | $F_{v_1}(t)$ corrected by linear regression using $T_s(t)$ |
| $F_{v_3}(t)$ | $F_{v_2}(t)$ corrected by linear regression using $(T_e(t) - T_s(t))$ |
| $\tilde{F}_v(t)$ | Total field corrected (final). |
| $\Delta F_0(t)$ | Difference between $F_v(t)$ and $F_s(t)$. |
| $\Delta F_{1-3}(t)$ | Differences between $F_{v_{1-3}}(t)$ and $F_s(t)$. |
| $(a,b)_{1-3}$ | Linear regression coefficients. |
| $H_{N0}$ | Baseline value of the horizontal component, calculated by the IGRF model. |
| $Z_0$ | Baseline value of the vertical component, calculated by the IGRF model. |





Because of the particular setup configuration of Macapá (Fig. 3), we got very different values for the temperature of the (GEOMAG) sensor and electronics (Fig. 10), and presumably these temperature variation influences on the Macapá data. We implement a methodology considering two types of correction in the data: diurnal variation (between GEOMAG and GSM - included in $\Delta F_0$) and temperature variation (included in $\Delta F_1$ and $\Delta F_2$), as shown in Fig. 11. The results demonstrate that there is a high correlation between the difference of GEOMAG and GSM with temperature of sensor, electronics and the difference between them (Fig. 12). Therefore, it is important to consider that variations of temperature may affect the data of magnetic stations and a correction may be important to be applied.

*Author contributions.* C.M. installed Macapá Station, performed the measurements and the computations. A.M. and J.M. planned Macapá station design. C.M., K.J.P. and J.M. drafted the manuscript. M.V.S. and R.P.C. designed the figures. All authors discussed the results and commented on the manuscript.

*Competing interests.* There are no financial or non-financial competing interests to report.

*Acknowledgements.* We would like to thank IEPA for the support on the Macapá station work, including the installation and availability of infrastructure and local staff. We also thank BioParque Amazônia for giving the space available to install the instruments and IMET for the ambient temperature data. KJP thanks to CNPq (Bolsa de Produtividade em Pesquisa - 306538/2022-9) for the support. We also thank the support within the funding programme "Open Access Publikationskosten" Deutsche Forschungsgemeinschaft (DFG, German Research Foundation) - Project Number 491075472





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
