# Peer review of "Macapá, a Brazilian equatorial magnetometer station: installation, data availability and methods for temperature correction."

_Geoscientific Instrumentation, Methods and Data Systems, 2023_

## Author Comment (AC1)

Review of the submission # **gi-2023-10 by Anatoly Soloviev**

**"Macapá, a Brazilian equatorial magnetometer station: installation, data availability and methods for temperature correction"**

by Cristiano Mendel Martins, Katia Jasbinschek Pinheiro, Achim Ohlert, Jürgen Matzka, Marcos Vinicius da Silva, and Reynerth Pereira da Costa

We thank Anatoly Soloviev for the helpful review.

Development of magnetic station network in addition to full-scale magnetic observatories increases the spatial resolution of the measurements and enables studying small-scale phenomena in the near Earth space. Therefore, the submission is definitely relevant and valuable for the community dealing with magnetic observations. Despite the generally positive impression from the work done by the authors, it is worth noting some aspects not covered in the manuscript:

1. The description of the pillar for the vector magnetometer is missing. Sensor shift in the course of the measurements can introduce significant distortions into the data. If the authors managed to build a compact non-magnetic pillar, this should be described in detail.

 We agree and add in line 68: '... pyramid described in Matzka et al. (2011). The foundation of the pyramid is made of concrete, about 25 cm thick and 70 cm deep. The instrument pillar is also from concrete. It is completely separate from the pyramid foundation, about 60 cm times 60 cm in cross section, 70 cm deep and extends about 30 cm above the ground level. The fluxgate sensor is located directly on the concrete pillar and surrounded and covered by a styrodur box with about 10 cm thickness. The GEOMAG electronics ...'

1. When solving the problem of the temperature correction of the magnetometer readings, it is necessary to describe the operation principle and technical specifications of the sensors provided by the manufacturer. Following this line, the discussion section lacks the comparison of the obtained temperature coefficients ($nT/C^0$) with the datasheet specifications.

We add after line 145: 'Now we can compare the observed temperature coefficients (typical values are $a2 = -0.3 \pm 0.15$ nT/∘C and $a3 = -0.04 \pm 0.025$) with the temperature coefficient given by the manufacturer. If sensor and electronics are operating at the same temperature, the observed temperature coefficient will be around $a2 = -0.3 \pm 0.15$ nT/∘C. This is close to the instrument specification of <0.2 nT/°C given by the manufacturer for the GEOMAG-02 fluxgate.

1. The temperature correction of the magnetometer recordings is routinely applied in the course of the geomagnetic data processing including the production of quasi-definitive and definitive data sets. There are some important publications devoted to

this issue. The authors should expand the relevant literature review and include the comparison with the existing approaches, for example:

- Janošek, M.; Butta, M.; Vlk, M.; Bayer, T. (2018). Improving Earth's Magnetic Field Measurements by Numerical Corrections of Thermal Drifts and Man-Made Disturbances. Sens., 2018, http://dx.doi.org/10.1155/2018/1804092
- Kudin D, Soloviev A, Matveev M, Shevaldysheva O. (2023). On a Novel Approach to Correcting Temperature Dependencies in Magnetic Observatory Data. Applied Sciences, 2023, 13(14):8008, https://doi.org/10.3390/app13148008

We agree and expand the introduction of the manuscript after line 55:

'Fluxgate magnetometers for measuring the components of the geomagnetic field are temperature sensitive and magnetometer stations typically do not have a temperature controlled environment. Several methods have been used and published to first determine temperature coefficients of such magnetometers and then correct the recorded raw data (e.g. Janosek et al, 2018; Kudin et al., 2023).

1. Multiple application of regressions might distort the data, as fluxgate type sensors have nonlinear temperature dependence. The authors apply regression three times over. It leads to both decreased correlation of the data adjusted step-by-step, and effect reduction when applying the correction. At the same time, after the first step of the correction the deltaF amplitude falls into the (-1, 1) interval, which satisfies high INTERMAGNET standards. Further corrections look somewhat redundant. To justify the repeated application of corrections, the authors should describe those scientific problems that set the requirements for data accuracy.

We fully agree with the reviewer that the deltaF obtained after the first correction step already yields very useful data that even can satisfy stringent INTERMAGNET standards. The motivation to continue is that the remaining deltaF variation shows an interesting structure, similar to the temperature variations, and we had an interest in identifying the reason for this.

1. The work results in obtaining adjusted deltaF and F derived from vector measurements. However, the next logical step is to obtain corrections for each component recorded by the vector magnetometer. The description of this important procedure is missing. This issue needs to be clarified.

We add after line 154:

'It would be preferable to determine a temperature coefficient for each of the three sensors of the Geomag 02. However, due to the location at the geomagnetic equator and because the instrument is oriented along magnetic north, the temporal change measured by the north sensor HN corresponds very closely to the temporal change measured by the scalar magnetometer. At the same time, both HE and Z0 + Z measure very small magnetic fields and hardly contribute to Fv. Therefore, the temperature coefficients determined here can be attributed to the HN sensor, while the temperature coefficients of the HE and Z sensor remain undetermined.'

1. Line 84 contains some glitch.

We correct the glitch.

The manuscript should be revised and resubmitted for review.

**Review by Jan Reda**
Comments on the research article gi-2023-10

Title: Macapá, a Brazilian equatorial magnetometer station: installation, data availability and methods for temperature correction

Authors: Cristiano Mendel Martins, Katia Jasbinschek Pinheiro, Achim Ohlert, Jürgen Matzka, Marcos Vinicius da Silva, and Reynerth Pereira da Costa

We thank Jan Reda for the helpful review.

GENERAL COMMENTS

The authors present the experiences related to the installation of a magnetic station near the geomagnetic equator in Northern Brazil. The place of installation in Macapá (MAA) is very interesting and important from the point of view of geomagnetic observations. As the authors notice, in this region of the globe, the fastest movement of the magnetic equator is observed, reaching up to 1100 km in 60 years. Such a location also allows observing the magnetic effect concerning the so called equatorial electrojet (EEJ). Especially valuable is the spatial and temporal observation of the electrojet phenomenon, performing magnetic analyses together with data from neighboring observatories.

The authors carefully selected the place for observation. They tested two locations: at Chaves and at IEPA (Institute for Scientific and Technological Research of the State of Amapá. The chosen location was optimal in terms of data quality, environmental disturbances, and logistical aspects of the observations.

Observing the EEJ effect on magnetic registration requires high-quality recording of changes in the components of the geomagnetic field. This primarily concerns reliable observations of diurnal variations, to which the authors pay particular attention because most vector magnetometers are sensitive to temperature changes, and diurnal temperature changes are particularly large in equatorial regions. Therefore, the authors developed a thermal correction method for the data from the vector magnetometer. The thermal correction method is a bit complicated, but the results of the correction are very good. It must be acknowledged that the authors clearly present how the improvement is after each of the multi-stage correction using linear regression methods. The Figure 11 clearly presents, which correction stages are more or less significant.

In summary, I am happy to recommend the manuscript for publication after making minor corrections listed below, in sections SPECIFIC COMMENTS and TECHNICAL, LANGUAGE AND OTHER REMARKS.

SPECIFIC COMMENTS

The Figure 12 is unclear (at least to me). The three plots in part a), the upper part, have the same horizontal axis description and a common vertical axis for all three, yet the correlation bars are different. Is this correct?

We correct Figure 12. Before, all panels were labeled with ΔF1, but they should read ΔF1 (left panel), ΔF2 (middle panel) and ΔF3 (right panel), we change the figure accordingly.

We also improve the figure caption to read 'Figure 12. Correlation of ΔF1 (left panel), ΔF2 (middle panel) and ΔF3 (top right panel) with sensor temperature Ts, electronics temperature Te and (Te-Ts) in blue in (a). RMSE (Root Mean Square Error) for each ΔF in t(b).'

TECHNICAL, LANGUAGE AND OTHER REMARKS
Line 33                     rather „Figure 1" instead of „figure 1"
Line 102                    "than" instead of "then"
Lines 130, 137        "datasets" instead of "dataset"
Line 199                    The name of the co-author is missing (B. Zhou)
Line 203                    The name of the co-author is missing (M. Mandea)
Line 207                    The name of the co-author is missing (E. Qamili)
Line 210                    The name of the co-author is missing (Jason J Green)
Line 212                    The name of the co-author is missing (Jürgen Matzka)
Line 214                    The name of the co-author is missing (Jürgen Matzka)
Line 217                    The name of the co-author is missing (Patrick Alken)
Line 220                    The name of the co-author is missing (C. Stolle)
Line 227                    It's worth adding DOI (https://doi.org/10.1007/s11214-016-0282-z)

We will address all these points in the revision.

ANNA WILLER

We thank Anna Willer for the helpful review.

**General comments:**

Overall, a well-written article on a new magnetometer station at an important location, especially important for monitoring the signal of the Equatorial Electrojet.

Detailed description of the setup including sensor orientation, schematic drawings, and instrument specifications. Well-described data corrections due to time and temperature issues.

I recommend the manuscript for publication. And have only minor comments and corrections described in the *Specific comments* and *Technical corrections.*

**Specific comments**

If I understood it correctly, there was not preformed any absolute measurements of the declination and inclination at the new magnetometer station (which is fine, as it is not a magnetic observatory). Instead, the baselines were determined by IGRF model values. It would be interesting to hear more on that procedure. I assume that the model parameters of the declination and inclination was used together with quiet variometer and scalar data?

In that case, I recommend that this is mentioned in section 4, where the baselines are introduced.

We add in line 115: '...the vertical sensor. Note that the east sensor has a negligible small offset which we set to 0 here. The baselines HN0 was obtained by subtracting a typical value of HN during nighttime and during quiet geomagnetic conditions from the simultaneous H value determined by the IGRF-13 model for the location of Macapá. The baselines Z0 was determined in analogue fashion. The difference...'

The horizontal north magnetic field in the sensor coordinate system $H_N$ is compared at different locations (for example in figure 8). The direction of $H_N$ is commonly determined by rotating the sensor such that $H_E$ is close to zero, as is described in the article. This means that the direction of $H_N$ can vary slightly between stations. The authors probably assume that the difference is so small that is has no effect in the data analysis preformed here. But perhaps it is worth mentioning that assumption somewhere in the text or figure caption?

**Technical corrections**

Section 2, Macapá Station Setup, page 3: Repeat of sentence: *Two instruments measuring the magnetic field were installed in Macapá.*

Section 2, Macapá Station Setup, page 4: GSM-90 overhauser by GemSYS, suggest to change to: GSM-90 Overhauser by GEM Systems

Section 3, Macapá Dataset, page 6: [label=.] ?

Section 3, Macapá Dataset, page 7: Figure 8 caption: repeat of the word "*the*": "Comparison between the *the* data…"

Section 4, Temperature correction, page 8: "…here denoted as Fs, measured by GSM-19" earlier it is mentioned that it was a GSM-90?

We correct in line 104: '..(here denoted as Fs, measured by the GSM-90, Table...'

Figure 6, 7a, 8d, and 9: I am assuming from the low values on the Y-axis that a mean value has been subtracted from F. In that case, I recommend that the authors briefly mention what has been removed from the data.

Section 4, Temperature correction, page 12: Figure 12a: The X-axis on two of the plots needs correction.

We make all the suggested changes.

Anonymous referee #4

We thank the anonymous referee for the helpful comments.

This manuscript addresses the nuts and bolts of installation of a magnetometer station in a specific location for specific purposes. The authors present a detailed study that includes a thorough description of technical tests, problems that turned up during the deployment, and the ways of their solutions, and, in many particular aspects, this manuscript can be regarded as some nuts and bolts of

I am really impressed with the presented results of the deployment of a new geomagnetic station which is undoubtedly going to improve the geomagnetic network and provide the data in particular for space weather studies. The principal advantage of the chosen location is its closeness to the geomagnetic equator, which provides a great monitoring opportunity.

However, the manuscript lacks some details that could be of reasonable interest for the researchers dealing with installation of magnetometers for long-term observations. In particular, it is not clear how the magnetometer was oriented in order to record the horizontal component: was a model value chosen as a reference, or were the absolute measurements carried out? I am convinced that the article should contain some details of this procedure before publication.

We agree and add in line

Technical corrections:

Line 67: "as it was done for Tristan da Cunha observatory Matzka et al. (2011)"

 - Should be "as it was done for Tristan da Cunha observatory (Matzka et al., 2011)" or "as it was done for Tristan da Cunha observatory as shown in (Matzka et al., 2011)"

We change to 'as it was done for Tristan da Cunha observatory (Matzka et al., 2011)'.